# Identification of Growth Patterns in Low Birth Weight Infants from Birth to 5 Years of Age: Nationwide Korean Cohort Study

**DOI:** 10.3390/ijerph18031206

**Published:** 2021-01-29

**Authors:** So Jin Yoon, Joohee Lim, Jung Ho Han, Jeong Eun Shin, Soon Min Lee, Ho Seon Eun, Min Soo Park, Kook In Park

**Affiliations:** Department of Pediatrics, Yonsei University College of Medicine, Seoul 06273, Korea; sojinyoon@yuhs.ac (S.J.Y.); imagine513@yuhs.ac (J.L.); feagd@yuhs.ac (J.H.H.); golden-week@yuhs.ac (J.E.S.); hseun@yuhs.ac (H.S.E.); minspark@yuhs.ac (M.S.P.); kipark@yuhs.ac (K.I.P.)

**Keywords:** developmental delay, child health, birth weight, growth measurement

## Abstract

This study aimed to investigate the nationwide growth pattern of infants in Korea according to the birth-weight group and to analyze the effect of growth on development. A total of 430,541 infants, born in 2013 and who received the infant health check-up regularly from 6 months to 60 months of age, were included. The weight, height, head circumferences percentiles, and neurodevelopment using screening tests results were compared among the birth-weight groups. Using longitudinal analysis, the study found a significant difference in height, weight, and head circumference, respectively, according to age at health check-up, birth weight group, and combination of age and birth weight (*p* < 0.001). The growth parameters at 60 months of age showed a significant correlation with those at 6 months of age especially in extremely low birth weight infants. The incidence of suspected developmental delay was significantly higher in infants with growth below the 10th percentiles than in those with growth above the 10th percentiles. Among 4571 (1.6%) infants with suspected developmental delay results at 60 months of age, birth weight, sex, and poor growth parameters were confirmed as associated factors. This nationwide Korean study shows that poor growth and neurodevelopment outcomes persisted among low-birth-weight infants at 60 months of age. Our findings provide guidance for developing a nationwide follow-up program for infants with perinatal risk factors in Korea.

## 1. Introduction

Due to the improvement in perinatal care, the survival and major morbidity free survival of preterm infants have improved dramatically. The focus of neonatology has shifted towards improving nutrition and anthropometry [1]. Growth and nutrition in preterm infants have long-term implications for neurodevelopmental and cardiometabolic outcomes [2]; consequently, growth monitoring is a cardinal precept of pediatric practice. 

A significant number of infants are discharged with their growth parameters still well below the normal range. In particular, very low birth weight (VLBW) infants and small for gestational age (SGA) preterm infants have a higher risk of growth deviations [3]. Several studies have shown an association between impaired extrauterine growth and poor long-term performance [4]. In moderate and late preterm children, poorer growth in the first seven years is associated with poorer neuropsychological functioning. Poor postnatal growth, especially head circumference, in preterm infants is associated with increased levels of motor and cognitive impairment [5]. 

The catch-up growth patterns of preterm infants have been a matter of debate. Approximately 80% of preterm infants after initial postnatal growth failure show catch-up growth in weight, length, and head circumference (HC), generally starting early in the first months of life and often achieving targets within the first two years of life [6]. However, late catch-up growth of preterm infants throughout childhood and even in adolescence has also been described. Catch-up growth is linked to an adverse health outcome, while rapid catch-up increases the risk of metabolic disease later in life [7]. 

In Korea, the total number of births in 2013 was 436,455, including 5.5% of which were low birth weight (LBW) infants and 0.7% of which were VLBW infants. The national health screening program in Korea checks anthropometric measurement and developmental progress serially until 6 years of age. However, little is known about the postnatal growth patterns of infants in Korea. Thus, it is important to use population-based nationwide data to understand of the early growth patterns of preterm infants. 

This study aimed to estimate the nationwide growth patterns according to the birth-weight group and to analyze the relationship between growth and development using a population-based surveillance system. We hope that our findings will inform policymakers, medical practitioners, and public health experts, and provide guidance for developing a nationwide follow-up program for public services, especially healthcare-delivery and social welfare delivery systems.

## 2. Materials and Methods

### 2.1. Patients and Data Source

We initially identified 430,541 infants who were born in 2013 and examined their infant health check-up records for the 1st to 6th visits from the National Health Insurance Service (NHIS) database. Healthcare claims including diagnostic codes of almost all Korean residents, approximately 98% covered by NHIS and 2% by medical aid, were linked to health check databases. The data, including gestational age and birth weight, were also grouped according to the International Classification of Diseases-10 codes (ICD-10: P07.01, P07.02, P07.09-14, P07.19, P07.20, P07.23, P07.29, P07.30, P07.39) [8]. The data were entered by the hospital or obtained from self-report questionnaires used by the national health screening program. Based on the birth statistics [9], the total number of births in 2013 was 436,455, and the number of infants who lived to be at least 1 year of age was 435,150, which shows that this study population 430,541 covered 99% of national births. 

The national health screening program for infants and children in Korea, launched in November 2007 is a kind of population surveillance system that consists of history taking, physical examination, anthropometric measurements, screening for visual acuity, and administration of Korean Developmental Screening Test (K-DST), oral examination, and questionnaires with anticipatory guidance [10]. The questionnaire contains the birth weight, preterm, vision, hearing, nutrition (meal, milk, snacks), multimedia, and safety education. We used only the information of birth weight and preterm status in questionnaire from family.

The period for national health screening program (1st to 6th visits) was divided and classified as follow; 6 months for 4–6 months of age, 12 months for 9–12 months of age, 24 months for 18–24 months of age, 36 months for 30–36 months of age, 48 months for 42–48 months of age, and 60 months for 54–60 months of age. The age at exam was defined as chronologic ages.

For growth assessment, the National health screening program checks anthropometric parameters including body weight, height, and HC serially at every follow-up. The percentile of growth was assessed using the Korean growth curve, which provides sex specific data. Poor growth was defined as measurements below the 10th percentile of weight, height, and head circumference individually.

The K-DST is used an effective screening tool for infants and children with neurodevelopmental disorders and has been used since 2011. It is used to verify whether infants are developmentally appropriate or neurodevelopmentally delayed in six domains: gross motor, fine motor, cognition, communication, social interaction, and self-control. 

The K-DST is conducted to screen children according to their corrected age before 36 months of age as recommendation and after that age, it is allowed to take tests according to chronological age. There is no K-DST at first visit, and at 5th, 6th visit (42–48, 54–60 months of age) the participants take the test papers according to their chronological age. The participants take the tests papers at the time of their clinic visit and get the result as four categorized groups based on the standard deviation (SD) scores; the scores above 1 SD are defined as ‘high-level’, those between −1 and 1 SD as ‘peer-level’, those between −2 and −1 SD as ‘follow-up test’, and those below −2 SD as ‘further evaluation’ [11]. Additional positive questions that take into account clinically important diseases, such as cerebral palsy, language delay, and autism spectrum disorders, that should be referred for ‘further evaluation’ are also included in the questionnaire. To evaluate the ability of the K-DST to identify infants with developmental delay, critical cutoff scores for 6 domains were set below −1 SD [12,13]. Suspected developmental delay was defined as a K-DST result of ‘further evaluation’ and ‘follow-up test’. 

In this study, growth and developmental results were analyzed according to five stratified birth weight groups (<1000 g, 1000–1499 g, 1500–1999 g, 2000–2499 g, and 2500–4500 g). LBW infants and VLBW infants were defined as having a birth weight below 2500 g and 1500 g, respectively. Preterm infants were defined as infants born before 37 weeks of gestation.

### 2.2. Statistical Analyses

The cohort was stratified according to the birth weight or the age of checkup. The characteristics of the subjects were expressed as means and standard deviations for continuous variables and as percentages for categorical variables. Correlations for height, weight and HC between 6 months and 60 months of ages as time periods were computed using Pearson’s correlation coefficient. Multiple logistic regression model was used to determine the independently associated factors with among infants with odds ratios (OR) and 95% confidence intervals (CI). Multivariate longitudinal data analysis was done using multivariate repeated measured model (PROC MIXED and GENMOD). All statistical analyses were performed using SAS version 9.4 (SAS Institute, Cary, NC, USA). *p*-values < 0.05 were considered statistically significant. 

### 2.3. Ethics Statement

In this study, all identifiable variables, including claim-, individual-, and organizational-level identification numbers, were re-generated in random by the NHIS database to protect the patients’ privacy. This study used NHIS data (NHIS-2019-1-569) maintained by the NHIS. The study protocol was approved by the Institutional Review Board of Gangnam Severance Hospital (No. 3-2019-0147). Informed consent was waived.

## 3. Results

### 3.1. Growth Outcome

Among 430,541 infants, born in 2013 and included in the study, 219,576 (51%) were male. The numbers of infants, who underwent health checks ranged from 286,331 (67%) to 347,153 (81%). The highest number of infants (*n* = 347,153, 81%) were included in the health check at 24 months of age. The highest number of preterm infants underwent the health check at 36 months (*n* = 26,338, 93%). The distribution of a number of infants who participated in the infant health check according to birth weight group were shown in Table 1. 

The mean percentile of weight, height, and HC according to age at health check was seen in Figure 1. Longitudinal analysis showed a significant difference in height, weight, and HC according to age, birth-weight group, and the combination of age and birth weight, respectively (*p* < 0.0001). The lower birth weight group showed a lower mean percentile of weight, height, and HC. There was a significant difference in height, weight, and HC between the low birth weight infants (<1000 g, 1000–1499 g, 1500–1999 g, 2000–2499 g) and the reference group with birth weight of 2500–4500 g according to age at health checkup. 

A total of 10,227 (7.4%) infants had a poor HC growth at 60 months of age, 10,950 (7.92%) infants had poor height growth, and 12,481 (9.03%) infants had a poor weight growth. Using longitudinal analysis, this study found a significant difference in the incidence of poor height, weight, and HC growth according to age at health check, birth-weight group, and combination of age and birth weight, respectively (*p* < 0.0001) (Figure 2). The lower birth-weight groups showed a higher incidence of poor weight, height, and HC growth. There was a significant difference in height, weight, and HC between the low birth weight group (<1000 g, 1000–1499 g, 1500–1999 g, and 2000–2499 g) and the reference group with birth weight of 2500–4500 g according to age at health checkup.

The Pearson correlation coefficient of the growth percentiles (the height, weight, and HC percentile) at 6 and 60 months of age obtained using correlation analysis is shown as Figure 3. Figure 3a was the results for the whole study population. Among the infants below 1000 g of birth weight, only weight showed a highly positive correlation (coefficient = 0.72) between 6 and 60 months of age, height (coefficient = 0.65) and HC (coefficient = 0.64) showed a moderate positive correlation. Pearson correlation coefficient analysis between 6 and 60 months of age was performed only among the infants who fell below 10th percentile in terms of height, weight, and HC at 60 months of age, and the results were shown in Figure 3b. Among the infants below 1000 g of birth weight, weight, height, and HC showed a weakly positive correlation; otherwise, there was no association. 

To analyze the relation between the growth at 6 months and 60 months, the risk of poor growth in the infants below 10th percentile of growth at 6 months was compared to the infants within 10–90th percentile of growth. The infant below the 10th percentile of HC and height at 6 months of age, respectively, showed the higher risk of HC and height below the 10th percentile at 60 months of age (HC, OR (95% CI) 1.62 (1.32–1.98); Height, 1.64 (1.38–1.95)). However, weight status at 6 months showed no significant association with the risk of weight below the 10th percentile at 60 months of age. 

### 3.2. Developmental Outcome

The incidence of suspected developmental delay result at 60 months of age was 10% (29,020). In particular, further evaluation was recommended for 4572 (1.5%) infants, and the ‘follow-up test’ for 24,448 (8.5%) infants. According to the birth-weight group, K-DST results were presented in Table 2. The smaller birth weight group had a greater number of ‘further evaluation’ results. The infants below 1000 g of birth weight were only 0.2% of the screened population, but among them, 14.8% of this group had ‘further evaluation’ and 21.0% of this group had ‘follow-up test’ recommendations. The lower birth-weight groups showed a higher incidence of suspected developmental delay. 

There is a significant difference in the incidence of suspected developmental delay results between the infants with poor weight, height, and HC growth and above 10th percentile at 60 months of age by birth weight group (Table 3). The infants with poor weight, height, and HC growth demonstrated higher frequency of suspected developmental delay results at 60 months of age. 

Lower birth weight, male sex, poor HC, poor height, and poor weight were confirmed as factors associated with suspected developmental delay results at 60 months of age using multivariate logistic regression analysis (Table 4). Infants with poor HC at 60 months of age had more suspected developmental delay results (OR 1.81, 95% CI 1.66–1.98), and the infants who weighed less than 1000 g at birth had more suspected developmental delay (OR 5.05, 95% CI 3.79–6.73) compared to infants with 2500–4500 g birth weight.

## 4. Discussion

This is the first large study showing the longitudinal growth and developmental patterns of children born with low birth weight in Korea. LBW infants are subject to a significant burden of morbidities, such as postnatal growth failure and neurodevelopmental impairments. However, extreme preterm infants have been the primary focus of the research over the years. In this study, longitudinal growth outcome in LBW infants from birth to 60 months was shown using nationwide population-based health check-up data. We confirmed an association between poor post-natal growth and developmental delay, both of which are persisting on long term follow-up, especially among LBW infants.

Our findings are consistent with those of previous international studies, which reported that a lot of preterm infants born lighter and shorter than full-term infants remain growth-restricted beyond the catch-up period [14]. We found that some degree of catch-up growth did occur with time; however, the difference remained until 60 months of age compared to the infants with 2500–4500 g. As shown in Figure 1, the smaller birth-weight group showed lower catch-up growth even at 60 months. Mean weight, height, and HC percentiles were persistently below 40 percent among LBW infants, as well as VLBW infants. Among children with poor growth, there is a decreasing trend in the incidence of poor growth until the 36 months of age, which then showed a stable or slightly increasing trend in the 48- and 60-months of age. Relatively poor growth can be seen in more preterm infants due to limitation of chronologic age up to 3 years of age; however, the difference in growth has persisted from 3 to 6 years old. Poor growth is still a serious problem in preterm infants, although there is an increase in survival and morbidity free survival in Korea. Therefore, close check-ups and support for catch-up growth until school age should be provided for preterm infants, as well.

Both the severity and duration of the growth retardation are related to the degree of prematurity of an infant [15]. We found the smaller the birth weight, the lower the mean growth percentiles, and the higher the incidence of poor growth. SGA children have a higher risk of growth failure throughout the follow-up period [16]. A 6-year follow-up study of very preterm infants showed the catch-up growth was mostly achieved before 2 months of age; however, it was continued until 6 years of age in SGA infants [17]. There have been reports of risk factors for growth failure that persist even after the catch-up period, and there are reports that SGA infants and more premature infants with morbidities are more vulnerable to growth failure [18,19,20]. In this study, SGA infants showed a higher risk for growth failure at 60 months of age than non-SGA infants did. In terms of height, most infants born SGA can catch-up by 2 years but around 15 % of them cannot achieve catch-up growth and remain short-heighted in adulthood. [19,21].

Children with poor growth have greater neurodevelopmental functioning problems than those with normal growth [22]. A nationwide Japanese population-based study analyzed the association of SGA infants with poor postnatal growth at 2 years of age with neurobehavioral development both at 5.5 and 8 years of age and reported that the findings warranted early detection and intervention for attention problems among these group [23]. Consistent with the other study, this research found that, children born with smaller birth weight showed poorer developmental results, and children with weight, height, HC less than 10th percentile at 60 months of age also showed higher incidence of the poor development, confirming growth is related to neurodevelopmental functioning.

Early postnatal growth is positively related to neurodevelopmental outcomes, especially Intelligence Quotient [22]; Postnatal growth rate of infants with intrauterine growth restriction has been associated with later cognitive outcomes, specifically Pylipow et al. reported that growth in the first 4 postnatal months is a risk factor for cognitive outcome at age 7 years [24]. Neurodevelopmental score at 8 year was related to weight, height, and HC at 8 years [25]. We confirmed the mean percentile of weight, height, and HC at 60 months of age were correlated with the mean growth percentile of 6 months of age. Therefore, close monitoring from early infant period and proper intervention for growth are important. 

Previous studies reported that growth restriction was more common in preterm infants but recent studies have shown positive reports of catch-up growth through nutritional support and quality improvement [26]. Small for gestational age infants with less than 28 week’s gestation had appropriate catch-up growth at term, improved with postnatal nutrition and care [27]. Early HC growth failure in very preterm infants can be improved by optimizing parenteral nutrition [28]. Although an aggressive nutritional strategy including using human milk fortifier or preterm formula, and high amino acid composition of parenteral nutrition were adopted in Korea, in this study we confirmed postnatal growth impairment is common in LBW infants, and catch-up growth may be delayed and incomplete in some. 

Children who fail to achieve catch up growth within 2 years of life remain short after childhood so an early initiation of growth hormone treatment was recommended by previous research [29,30]. The length Z and changes of scores at 12 months of corrected age may be correlated with catch-up height at 3 years and so it is useful for earlier initiation of growth hormone treatment in VLBW infants [31]. We found that the infants with a height below 10th percentile at 60 months of age were more numerous in VLBW (25%) group than in LBW (14%) or 2500–4500 g (5%) group. At 60 months of age, the mean percentile of height had correlations with the mean height percentile of 6 months of age.

The main strength of this study was that it was a nationwide study with a large population and it was able to report an association between growth and neurodevelopmental outcomes overtime. A total of 99% of eligible infants participated in the national health screening program for infants and children, so these results are an accurate representation of the growth of the infant population in Korea. These nationwide data accounts for all infants, including LBW but not VLBW infants, who participated in health check-ups during the first five years of life in addition to the infants weighing 2500–4500 g infants as a reference.

There are some limitations to this study. Since weight is the most important factor in the growth assessment of newborns and infants, this study design was analyzed based only on birth weight. Because birth weight is usually related with gestation, and SGA status can make some discrepancy, we just display the infants dividing birth weight rather than gestational age. Poor growth was defined as weight, height, and HC individually below the 10th percentile. VLBW infants accounted for 0.7% of total birth infants in this cohort, which is a very small proportion. Preterm infants with intraventricular hemorrhage or post-hemorrhagic hydrocephalus have larger HC but the effect isn’t considered due to a small population. There are many factors having affecting on growth including nutrition and co-morbidities, but we lack detailed data that may provide information on important confounders. For the integrity of data on growth in preterm infants, we included the growth parameter at only postnatal age, but not the corrected age.

## 5. Conclusions

This Korean population-based study showed that a significant number of LBW infants did not achieve catch up growth even at 60 months of age. Close monitoring of appropriate weight gain, nutritional intervention, and early intervention programs will be needed for improving children’s growth and developmental outcomes. Our findings provide guidance for developing a nationwide follow-up program for infants with perinatal risk factors.

## Figures and Tables

**Figure 1 ijerph-18-01206-f001:**
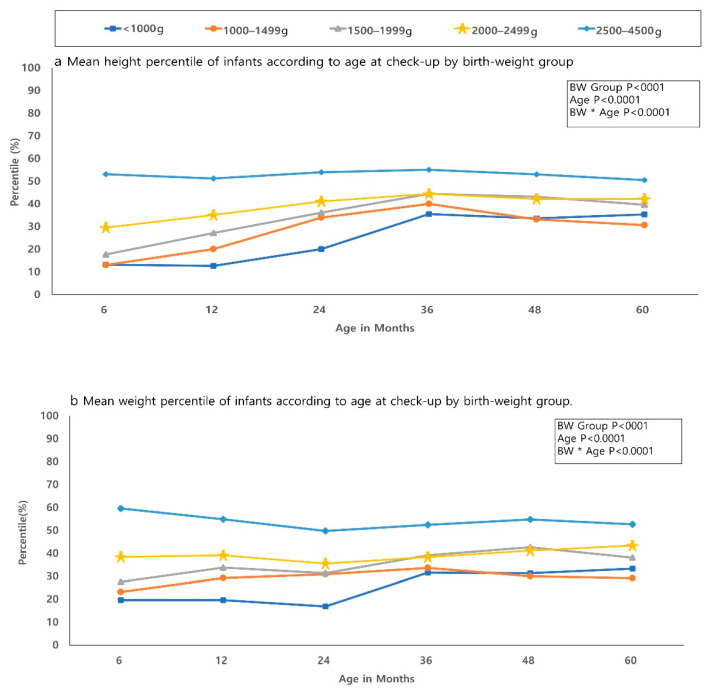
Mean growth percentile of infants according to age at check-up by birth-weight group. (**a**) Height. (**b**) Weight. (**c**) Head circumference (HC). Significant differences in height, weight, and HC between the low birth weight infants groups and the reference group according to age at health checkup were shown. *p*-values were significant in height, weight, and head circumference between each low birth weight group and the reference group compared at age of health check-up.

**Figure 2 ijerph-18-01206-f002:**
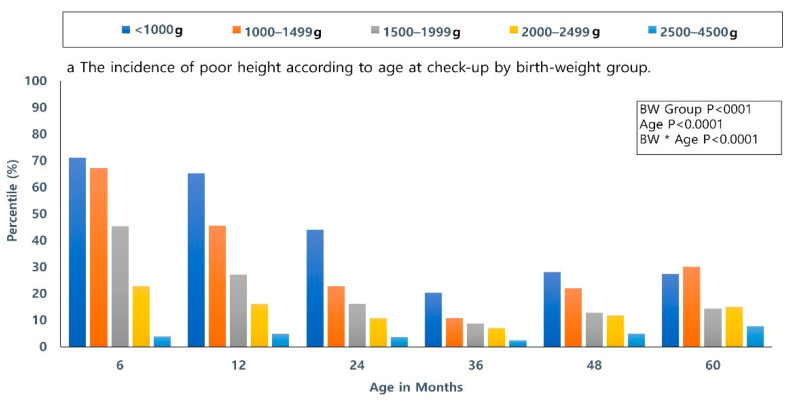
The incidence of poor growth (below 10th percentile) according to age at check-up by birth-weight group. (**a**) Height. (**b**) Weight. (**c**) Head circumference. A higher incidence of poor weight, height, and HC growth in the lower birth-weight groups was noted. A significant difference in height, weight, and HC between the low-birth-weight groups and the reference group according to age at health checkup were shown. *p*-values were significant in height, weight, and head circumference between each low-birth-weight group and the reference group compared at age of health check-up.

**Figure 3 ijerph-18-01206-f003:**
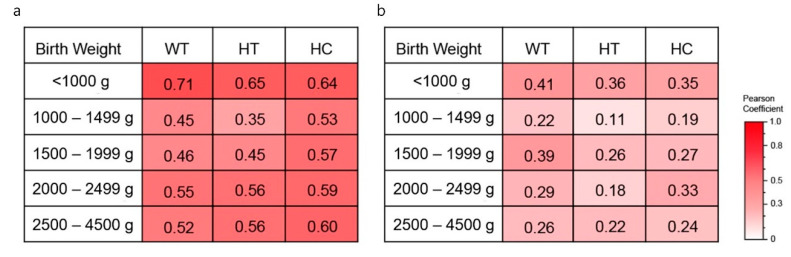
Pearson correlation coefficient between 6 months of age and 60 months of age for birth-weight groups among whole population (**a**) and among infants who were below the 10th percentile of height, weight, and HC at 60 months of age (**b**).

**Table 1 ijerph-18-01206-t001:** The population characteristics at the health checkup according to birth-weight group.

Age at Exam	Total Infants	<1000 g	1000–1499 g	1500–1999 g	2000–2499 g	2500–4500 g	Preterm
6 months	311,446 (72.3)	137 (0.0)	693 (0.2)	2332 (0.7)	11,539 (3.7)	295,989 (95.0)	11,398 (3.7)
12 months	313,235 (72.8)	196 (0.1)	819 (0.3)	2487 (0.8)	11,826 (3.8)	297,151 (94.9)	10,919 (3.5)
24 months	347,153 (80.6)	314 (0.1)	1376 (0.4)	3050 (0.9)	13,320 (3.8)	328,176 (94.5)	12,355 (3.6)
36 months	344,468 (80.0)	643 (0.2)	1959 (0.6)	3495 (1.0)	13,240 (3.8)	324,086 (94.1)	26,338 (7.6)
48 months	323,958 (75.2)	606 (0.2)	1425 (0.4)	3747 (1.2)	12,809 (4.0)	304,361 (94.0)	26,028 (8.0)
60 months	286,331 (66.5)	662 (0.2)	1174 (0.4)	4584 (1.6)	12,288 (4.3)	266,669 (93.1)	23,542 (8.2)

Data are presented as Number (%).

**Table 2 ijerph-18-01206-t002:** The results of developmental screening test at 60 months of age according to birth weight group.

BW Group	Number of Infants	Further Evaluation							Follow-Up Test							Peer & High Level
		Total	Gross Motor	Fine Motor	Cognition	Communication	Social Interaction	Self-Control	Total	Gros Motor	Fine Motor	Cognition	Communication	Social Interaction	Self-Control	
<1000 g	529	56 (11)	48	48	45	47	39	42	91 (17)	59	47	44	40	25	30	382 (72)
1000–1499 g	1067	63 (6)	56	53	46	48	45	46	142 (13)	65	62	54	57	35	53	862 (81)
1500–1999 g	2316	96 (4)	68	64	65	72	59	56	285 (12)	111	123	104	95	75	93	1935 (84)
2000–2499 g	9381	254 (3)	143	168	184	198	156	127	947 (10)	332	383	369	342	269	293	8180 (87)
2500–4500 g	263,579	4094 (2)	2201	2603	2825	3089	2454	2111	22,930 (9)	7394	8234	8848	8820	5938	7013	236,555 (89)

Data are presented as Number or Number (%). BW, birth weight.

**Table 3 ijerph-18-01206-t003:** Poor growth outcomes (below 10th percentile) at 60 months of age according to birth-weight group among the infants with suspected developmental delay results.

BW Group	Number of Infants	WT Poor	HT Poor	HC Poor
<1000 g	147	96 (65)	73 (50)	86 (59)
1000–1499 g	205	71 (35)	88 (43)	83 (40)
1500–1999 g	381	112 (29)	117 (31)	146 (38)
2000–2499 g	1201	319 (27)	342 (28)	393 (33)
2500–4500 g	27,024	2520 (9)	2663 (10)	2958 (11)

Data are presented as No. (%). BW, birth weight; WT, weight; HT, height; HC, head circumference.

**Table 4 ijerph-18-01206-t004:** Multivariate logistic regression analysis for the suspected development delay results at 60 months of age.

Variable	OR (95% CI)	*p* Value
Poor HC at 60 months of age	1.81 (1.66–1.98)	<0.001
Poor HT at 60 months of age	1.63 (1.48–1.80)	<0.001
Poor WT at 60 months of age	1.18 (1.07–1.30)	<0.001
BW 1000 g vs. 2500–4500 g	5.05 (3.79–6.73)	<0.001
1000–1499 g vs. 2500–4500 g	3.05 (2.35–3.96)	
1500–1999 g vs. 2500–4500 g	2.37 (1.92–2.92)	
2000–2499 g vs. 2500–4500 g	1.64 (1.44–1.87)	
Male vs. Female	2.02 (1.90–2.15)	<0.001

HC, head circumference; HT, height; WT, weight; BW, birth weight.; OR, odds ratio; CI, confidence interval.

## Data Availability

Not applicable.

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
