# Peer review of "Identification of Growth Patterns in Low Birth Weight Infants from Birth to 5 Years of Age: Nationwide Korean Cohort Study"

_ijerph, 2021, doi:10.3390/ijerph18031206_

Round 1

Reviewer 1 Report

I would like to congratulate you for conducting a research of great interest, that is also methodologically excellent. My only concerns is in the section of "Materials and Methods" for further clarification on how you defined ''poor developmental''.

Author Response

We are grateful for the reviewer’s thoughtful comments and giving us the opportunity to improve our manuscript by additionally clarifying the definition of "poor developmental" in our revised manuscript.  

  • The Korean Developmental Screening Test for Infants and Children (K-DST) is reliable and valid, suggesting its good potential as an effective screening tool for infants and children with neurodevelopmental disorders in Korea since 2011. Based on the previously analyzed standard deviation scores, the scores above 1 standard deviation (SD) were defined as ‘high-level’, between -1 and 1 SD as ‘peer-level’, between -2 and -1 SD as ‘follow-up evaluation, and below -2 SD as ‘further evaluation’.
  • Several studies reported the validity of KDST by distinguishing between normal and neurodevelopmentally delayed groups. To evaluate the ability of the K-DST to identify infants with developmental delay, critical cutoff scores for 6 domains (gross motor, fine motor, cognition, language, sociality, and self-help) were set below -1 standard deviation (SD) such as ‘follow-up or further evaluation’. The K-DST had a high discriminatory ability with a sensitivity of 0.833 and specificity of 0.979. The language and cognition domain of the revised K-DST was highly correlated with the K-Bayley Scales of Infant Development-II’s Mental Age Quotient (r=0.766, 0.739), while the gross and fine motor domains were highly correlated with Motor Age Quotient (r= 0.695, 0.668), respectively. The Verbal Intelligence Quotient of Korean Wechsler Preschool and Primary Scales of Intelligence was highly correlated with the K-DST cognition and language domains (r=0.701, 0.770), as was the performance intelligence quotient with the fine motor domain (r=0.700_
  • We described “poor development” as poor developmental screening results that included scores below -1 SD as the further evaluation and the follow-up test.
  • Another reviewer recommended “poor development” to change “suspected developmental delay”.

So we described it in the method as the Suspected developmental delay was included as the further evaluation and the follow-up evaluation.

Reference> Jang CH, Kim SW, Jeon HR, Jung DW, Cho HE, Kim JY, and Lee JW. Clinical Usefulness of the Korean Developmental Screening Test (K-DST) for Developmental Delays. Ann Rehabil Med. 2019,43(4): 490–496.

Yim CH, Kim GH, Eun BL. The usefulness of the Korean Developmental Screening Test for infants and children for the evaluation of developmental delay in Korean infants and children: a single-center study. Korean J Pediatr. 2017,60(10):312-319

Chung HJ, Yang D, Kim SK, Kim SW, Kim YA etc. Development of the Korean Developmental Screening Test for Infants and Children (K-DST) CEP 2020,Vol. 63(11), 438–446.

Kim CY, Jung E, Lee BS, Kim KS, Kim Ellen AR. Validity of the Korean Developmental Screening Test for very-low-birth-weight infants. Korean J Pediatr. 2019,62(5):186-192

Reviewer 2 Report

 Identification of growth pattern in low birth weight infants from Birth to 5 Years of Age: Nationwide cohort study

Summary

National, longitudinal cohort study of growth and development among low birth weight infants from 6 months till 6 years of age. Involving around 430, 500 infants, authors concluded that poor growth persisted among the low birth weight infants at 6 years of age. Among extremely LBW infants, the correlation was significant between 6 and 60 months.

General comments – came across many typos and grammatical errors. Though English may not be the first language for the authors, these need to be addressed by proof reading or by readers proficient in English, which will make this manuscript even better. The focus of this review is on the main content though I tried to correct some sentences and words

Title – to be corrected as ‘study’. Also suggest ‘growth patterns ‘.

Abstract

Over all, written well.

Line 24-26 – conclusion – reads better if corrected to something like; poor growth and neuro development persisted among low birth weight infants on long term follow up

Introduction /background

Introduction and justification for the study/ knowledge gap are appropriate. These type of studies based on representing population will be useful for policy making and implementing relevant programs as mentioned by the authors.

It may be useful to provide some baseline statistics about total number of births every year, rates of LBW/SGA etc in this population

Line 34 – instead of infants discharged with growth restriction, reads well if re written as ‘ a significant number of infants are discharged with their growth parameters still well below the normal range.

Methods

This section needs some definitions and restructuring to avoid confusion.  LBW, VLBW should be defined. Preterm infants were also included in the results ( table 1) This should be defined as < 37 weeks and also clarify the growth parameters were taken at corrected ages rather than chronological age.

Authors used the term ‘poor developmental result ‘ to describe neuro development based on K-DST. Is this a standard term to describe neuro development? if not suggest changing to either ‘ developmental delay or  developmental impairment  or suspect or confirmed developmental delay/impairment

Suggest to use one paragraph to describe growth related assessments, definitions and follow up. Similarly a second paragraph to describe the ‘development’ part including K-DST, definitions and follow ups.

Line 61 – inputted by ‘can be changed to entered/provided by

Statistics - no suggestions

Results

If there is no limit on number of tables, please include a table with details of K-DST results – like how many belonged to each of the 4 groups according to results. At least include as supplement

Fig 1 – noted that none of the growth curves were above 50th centile including good birth weight groups, it is almost flat, also none of the catch up curves never reached more than 40 centile, and they are all flat. It is slightly concerning. Authors need to explain this in discussion. Is this because different number of babies were followed up at different times?  

Fig 3 – need to explain A and B in the legend which are 2 different assessments /correlations

Discussion

Lines 164-165extreme preterm has been the primary focus of the research over the years.

Lines 167-168 – we confirmed an association between poor post-natal growth and developmental delay , both of them are persisting on long term follow up especially among low birth weight infants

Lines 169 – our findings are consistent with previous studies

Line 170 – remain growth restricted beyond the catch up period

Lines 171- 172 – catch up growth has occurred ( not has been occured ), as the time goes on , however the difference remained ( not has been remained)

Line 214 – within 2 years

Line  221 – infants participated

Line 224 – included VLBW infants ( not clear what is the meaning of this sentence)

Limitations – note that the number of infants who were ELBW and VLBW was small in this cohort to come to a meaningful conclusion though they are the most high risk group for growth and neuro developmental delay. This should be acknowledged

Conclusions

Line 231 – 232 – this should be revised – did not reach ( not reached)  – A significant number of  LBW infants did not achieve catch up growth even at 60 months of age.

References  - I assume these are according to the journal requirement. Please look at references 9, 10 and 12 if they need revision

Reviewer 3 Report

The overall concept of this paper is good, however, there are multiple typos as well as serious flaws in how data is presented.

Abstract - the authors state that certain characteristics (ie birth weight, sex, etc) are confirmed as risk factors for poor development.  However, this was an observational study, so these are associations, not risk factors.

Introduction section

In the introduction section, the authors state that "the focus of neonatology has shifted toward improving nutrition and anthropometry."  What did the focus used to be?

Many of the references in the introduction do no match the information being presented.  These need to be evaluated and re-done.

Materials and methods

  • Patient data
    • The number of infants born in 2013 is less than the number alive at 1 year of age. 
    • There is no discussion in the materials/methods of the weight groups (this is not presented until the results section). 
    • The authors do not discuss why the use the data they use.  Why are they stratifying based on birth weight but not presenting birth data in their charts? (weight, height, head circumference)
    • They discuss using ICD-10 codes, however, do not state what codes they reviewed. 
    • They reference a questionnaire given to the families but do not present what the questionnaire contains.
    • The reference for the K-DST is in Korean and the abstract for it only discusses needing something better than the K-ASQ.
    • Information about the time points for follow up visits is presented in different portions of this section (leaves you confused while reading). 
    • Poor growth is defined in this section as below the 10th percentile of weight, height and head circumference.  Do they really mean all 3 have be below the 10th percentile? Some babies with small growth have head sparing. Also, premature babies with brain bleeds and post-hemorrhagic hydrocephalus end up with large heads but are often still below goal weight or height.
  • Statistical analysis
    • State the cohort was stratified according to birth weight or the age at check-up. - this doesn't make sense and may just be a typo
    • The authors mention using one-way ANOVA or chi-square test, however, this analysis is not presented anywhere in the paper.

Results

  • There is no chart with patient data in the results section as typically seen in many other papers with range of gestational ages, # of each gender, etc.  In fact, there is no information throughout the paper about gestational age at birth.
  • The authors reference a supplemental table - there is no supplemental table provided.
  • The authors continue to say the "control" weight group is 2000-2400g, however, their tables present the control as 2500-4500g
  • Are growth parameters based on corrected gestational age or just post-natal age? Growth parameters can be very different for a baby born at 24 weeks vs 36 weeks and premature babies are not expected to fully catch up until 2 years of age.
  • Figure 2 - which of these have significant p-values? Most charts like this have a star to mark what is significant. This graph just presents numbers and says that the p-value is <0.001. 
  • The description of the Pearson Correlation Coefficient data does not make a lot of sense.  This entire paragraph needs to be re-written/explained better.  Line 128-133 the authors discuss information that is not presented in a chart/graph.
  • Developmental outcome - this section is very short and needs to be expanded upon further.  While growth is a concern for physicians, developmental outcome is more concerning as it has a bigger impact during school and beyond.

The entire discussion section does not flow well and needs to be rewritten.  The authors will reference a study and then in the next sentence state "in this study." Do they mean in the reference presented or in the current study they performed?

Round 2

Reviewer 3 Report

I appreciate the authors responses to previous comments, however, not all responses were not added to the paper. 

  • The number of infants included in the study vs the number of infants born and alive at one year of age still don't match. Authors need to include why these numbers are different IN the paper).
  • ICD 10 codes used are not listed.
  • Data obtained from the questionnaire is not included in the paper.

Reference 6 does not match data reference in the introduction.

Line 84 - patients should not referred to as "normal", instead try "developmentally appropriate" or something to that effect.

Overall, the authors need to decide if they want to discuss preterm infants or term infants.  The introduction and much of the conclusion talks about preterm data, however, the data presented does not separate out preterm and term data. If preterm infants are going to be included in the study and presented, this should be separated out because they probably skew the rest of the data.  I'd be very interested to see growth and development of term SGA infants.

Figures and their descriptions should stand alone. The authors need to expand on p-value data in the descriptions of figures (not just in the results section).

Growth parameters are at chronologic age but developmental screenings are at corrected age - need to explain why or include that developmental screens are at corrected age in limitations.

Line 232-234 - not really sure what the authors are trying to say here.
Line 245 - not sure what the authors are saying here.

Author Response

Response to Reviewer 3 Comments

  • We are grateful for the reviewer’s thoughtful comments and giving us the opportunity to improve our manuscript. We have addressed and better clarified the following points in detail. Also, amendments were reflected by requesting English editing service.

Point 1: The number of infants included in the study vs the number of infants born and alive at one year of age still don't match. Authors need to include why these numbers are different IN the paper).

Response 1:  Thank you for the thoughtful comment. I rewrote the number of  study population in the paragraph. The total number of births in 2013 was 436,455 and the number of infants who lived to be at least 1 year of age was 435,150, which shows that this study population 430,541 covered 99% of national births.

Point 2: ICD 10 codes used are not listed.

Response 2: Thank you for the good point. I added ICD 10 codes in the data source paragraph : International Classification of Diseases-10 codes (ICD-10: P07.01, P07.02, P07.09-14, P07.19, P07.20, P07.23, P07.29, P07.30, P07.39) .

Point 3: Data obtained from the questionnaire is not included in the paper.

Response 3: We appreciate the reviewer’s thoughtful comment. We added description about questionnaire in the paper: ‘The questionnaire contains the birth weight, preterm, vision, hearing, nutrition (meal, milk, snacks), multimedia, and safety education. We used only the information of birth weight and preterm status in questionnaire from family’

Point 4: Reference 6 does not match data reference in the introduction.

Response 4: There is a typo error during editing. Deleted that reference.

Point 5: Line 84 - patients should not referred to as "normal", instead try "developmentally appropriate" or something to that effect.

Response 5: As recommended, we have corrected “normal” as “developmentally appropriate”.

Point 6: Overall, the authors need to decide if they want to discuss preterm infants or term infants.  The introduction and much of the conclusion talks about preterm data, however, the data presented does not separate out preterm and term data. If preterm infants are going to be included in the study and presented, this should be separated out because they probably skew the rest of the data.  I'd be very interested to see growth and development of term SGA infants.

Response 6: We thank the reviewer for helpful comments. In limitation, we added the sentence:” Since weight is the most important factor in the growth assessment of newborns and infants, this study design was analyzed based only on birth weight. Because birth weight is usually related with gestation, and SGA status can make some discrepancy we just display the infants dividing birth weight rather than gestational age.”

Point 7: Figures and their descriptions should stand alone. The authors need to expand on p-value data in the descriptions of figures (not just in the results section).

Response 7: Thank you for the good point. I added sentence in Figure 1 leagend: Significant differences in height, weight, and HC between the low birth weight infants groups and the reference group according to age at health checkup were shown.

I added sentence in Figure 2 leagend: A higher incidence of poor weight, height, and HC growth in the lower birth-weight groups was noted. A significant difference in height, weight, and HC between the low birth weight groups and the reference group according to age at health checkup were shown.

Point 8: Growth parameters are at chronologic age but developmental screenings are at corrected age - need to explain why or include that developmental screens are at corrected age in limitations.

Response 8: We thank the reviewer to point out the importance of screening age. We added some explanation in ‘methods’  paragraph. The K-DST is conducted to screen children according to their corrected age before 36 months of age as recommendation and after that age, it is allowed to take tests according to chronological age. There is no K-DST at first visit, and at 5th, 6th visit (42-48, 54-60 months of age) the participants take the test papers according to their chronological age.

Point 9: Line 232-234 - not really sure what the authors are trying to say here.

Point 10: Line 245 - not sure what the authors are saying here.

Response  9, 10  :  Thank you for the thoughtful comment. We revised the discussions including line 232-245. We discussed the relationship between poor growth in childhood and the outcome of poor neurodevelopment in childhood.  Next, we described the link between early sluggish growth and late neurodevelopmental problems.
